# Insulin Resistance Is Cheerfully Hitched with Hypertension

**DOI:** 10.3390/life12040564

**Published:** 2022-04-10

**Authors:** Susmita Sinha, Mainul Haque

**Affiliations:** 1Department of Physiology, Khulna City Medical College and Hospital, 33 KDA Avenue, Hotel Royal Mor, Khulna Sadar, Khulna 9100, Bangladesh; sinhasusmita24@gmail.com; 2The Unit of Pharmacology, Faculty of Medicine and Defence Health, Universiti Pertahanan Nasional Malaysia (National Defence University of Malaysia), Kem Perdana Sungai Besi, Kuala Lumpur 57000, Malaysia

**Keywords:** high blood pressure, resistance to insulin action, type 2 diabetes, endothelial dysfunction

## Abstract

Cardiovascular diseases and type 2 diabetes mellitus (T2DM) have risen steadily worldwide, particularly in low-income and developing countries. In the last hundred years, deaths caused by cardiovascular diseases increased rapidly to 35–40%, becoming the most common cause of mortality worldwide. Cardiovascular disease is the leading cause of morbidity and mortality in type 2 diabetes mellitus (T2DM), which is aggravated by hypertension. Hypertension and diabetes are closely interlinked since they have similar risk factors such as endothelial dysfunction, vascular inflammation, arterial remodeling, atherosclerosis, dyslipidemia, and obesity. Patients with high blood pressure often show insulin resistance and have a higher risk of developing diabetes than normotensive individuals. It has been observed that over the last 30 years, the prevalence of insulin resistance (IR) has increased significantly. Accordingly, hypertension and insulin resistance are strongly related to an increased risk of impaired glucose tolerance, diabetes, cardiovascular diseases (CVD), and endocrine disorders. Common mechanisms, for instance, upregulation of the renin–angiotensin–aldosterone system, oxidative stress, inflammation, and activation of the immune system, possibly have a role in the association between diabetes and hypertension. Altogether these abnormalities significantly increase the risk of developing type 2 diabetes.

## 1. Introduction

The occurrence of high blood pressure is a risk factor for type 2 diabetes mellitus and accelerates the development of insulin resistance [1]. Hypertension and insulin resistance are strongly related to an increased risk of impaired glucose tolerance, diabetes, cardiovascular diseases (CVD), and endocrine disorders [2]. The European Society of Cardiology (ESC) and the European Society of Hypertension (ESH) published their hypertension guideline in 2018 [3]. It is recommended that subjects with blood pressure greater than 140/90 mmHg, or on medications for hypertension are defined as having hypertension. Again, pre-hypertension was determined if blood pressure was 120–139/80–89 mmHg in subjects without medicines for hypertension [4]. Hypertension is a risk factor for diabetes and can lead to diabetes [5]. According to American Diabetes Association (ADA) guidelines, diabetes mellitus can be defined as fasting plasma glucose ≥ 126 mg/dL, oral glucose tolerance test (OGTT) 2 h plasma glucose ≥ 200 mg/dL, and HbA1C ≥ 6.5%, or if anti-diabetic medications were used [6]. Recent studies suggested that the rapid progression of insulin resistance may be associated with an increased risk of diabetes. Furthermore, insulin resistance has been proposed to be the underlying cause linking hypertension and diabetes. Theoretically, these mechanisms may persist over time and could accelerate the progression of insulin resistance in hypertensive subjects [7,8]. 

### 1.1. The Objective of This Study

This review provides insights to evaluate the impact of hypertension, the relationship of increased blood pressure to the risk of type 2 diabetes, and insulin resistance progression. 

### 1.2. Materials and Methods

This review attempts to describe the possible mechanisms related to insulin insensitivity, type 2 diabetes, and hypertension. The literature search was based on electronic databases and carried out using Google search engine, Google Scholar, and PubMed. Related articles from the list of references were also searched to obtain more articles on the topics above. Keywords used in search of associated articles were “Insulin resistance”, “Hypertension”, “Insulin Resistance and Hypertension”, “End organ damage in hypertension with insulin resistance”, and “Possible mechanisms linking insulin resistance and hypertension and progression of Type 2 Diabetes”.

### 1.3. Insulin and Its Effects

Insulin first binds with and activates a membrane receptor protein and causes subsequent effects. The insulin receptor combines four subunits held together by disulfide linkages, i.e., two alpha and two beta subunits. Insulin binds with the alpha subunits outside the cell membrane; as linkages with the β subunits, the cell becomes autophosphorylated [9]. This autophosphorylation of the beta subunits of the receptor activates tyrosine kinase, which in turn causes phosphorylation of multiple other intracellular enzymes called insulin receptor substrate (IRS) [10]. Different types of IRS are expressed in various tissues. Thus, insulin leads the intracellular mechanism to produce the preferred effects on carbohydrate, fat, and protein metabolism [11]. Insulin maintains glucose homeostasis by integrating carbohydrates, protein, and lipid metabolism in healthy individuals [12]. Insulin increases glucose uptake in muscle and liver and inhibits lipolysis and hepatic gluconeogenesis [13,14]. Loss of sensitivity to insulin action contributes to hypertension due to the loss of the vasodilator effect of insulin and vasoconstriction caused by FFAs [15]. Reduction in insulin-mediated glucose disposal leads to compensatory hypersecretion of insulin to maintain homeostasis, and glucose intolerance results if the endocrine pancreas response is insufficient [16]. Several common risk factors, such as genetic factors, obesity, dyslipidemia, and insulin resistance, underlie the pathophysiological relationship between hypertension and diabetes mellitus. In addition to this, insulin resistance is the critical mechanism that connects these conditions (Figure 1) [17].

## 2. Type 2 Diabetes Mellitus and Hypertension

It is predicted that the number of cases of T2DM will increase from 415 million to 642 million by the year 2040 [18]. Moreover, hypertension affects most individuals with diabetes mellitus. Co-occurrence of diabetes mellitus and hypertension can increase the risk of morbidity and mortality from cardiovascular disease. Essential hypertension is characterized by both hemodynamic and metabolic abnormalities [19]. Epidemiological studies have documented a high incidence of diabetes in hypertensive patients [20]. The evidence reports a strong correlation between hypertension and T2DM [21]. Patients with essential hypertension and concomitant hypertension-related target organ damage, such as left ventricular hypertrophy carotid atherosclerosis, are at higher risk of incidence of T2DM in comparison to those without targeted organ damage [22]. It is not a coincidence that hypertension and diabetes can frequently coexist in the same individual [23], as they have shared mechanisms, such as upregulation of the renin–angiotensin system, oxidative stress, inflammation, and activation of the immune system, also common pathophysiological aspects [24], primarily those related to obesity and insulin resistance.

Diabetes is associated with macrovascular (includes large arteries such as conduit) and microvascular (includes small arteries and capillaries) disease [25]. Chronic hyperglycemia and insulin resistance have an important role in the initiation of vascular complications of diabetes consisting in several mechanisms [26], such as (1) increased formation of advanced glycation end products (AGEs) and activation of the receptor for advanced glycation end products (RAGE) AGE–RAGE axis, (2) oxidative stress, and (3) inflammation [27]. Furthermore, emerging evidence suggests a role for microRNAs (miRNAs) in the vasculopathy of diabetes [28]. Hypertension is a significant risk factor for diabetes-associated vascular complications because hypertension itself is characterized by vascular dysfunction and injury [29].

### 2.1. Macrovascular Diseases in Diabetes

Macrovascular or cardiovascular disease is a complex inflammatory process that leads to myocardial infarction, stroke, and peripheral artery disease. Individuals with diabetes or prediabetes have a higher risk of developing these complications [30,31]. Atherosclerosis is the primary pathologic process linked to macrovascular disease, and extensive distribution of vascular lesions accelerates the development of diabetes. Hyperglycemia and insulin resistance contribute to atherosclerotic changes and the pathogenesis of macrovascular complications in diabetes mellitus [32].

#### Pathophysiological Features of Macrovascular Disease

It has been observed that insulin resistance is detectable for several years before the onset of T2DM [33]. In addition, insulin resistance can be associated with obesity, mainly central obesity [34]. Adipocytes in both subcutaneous ad visceral areas, undergo hypertrophy during calorie excess, in obese humans, [35,36]. Moreover, visceral adipocytes are more susceptible to cell death as they enlarge, and their stromal vascular fraction infiltrates macrophages [37]. Then, these macrophages around dead adipocytes form crown-like structures associated with cytokine [38], including tumor necrosis factor-alpha (TNF-α), interleukin-6 (IL-6), and inducible nitric oxide synthase [39]. These changes are consistent with the onset of insulin resistance and deliver a pathophysiological relationship between metabolic and vascular disease.

These functional changes and associated low-grade inflammation in endothelial and smooth muscle cells of the vascular wall cause cell proliferation, hypertrophy, remodeling, and apoptosis [40]. This accelerates disturbance of the balance between the arterial wall scaffolding proteins, specifically elastin and collagen; these proteins determine vascular compliance and are recognized as a form of vascular aging [41,42]. In cardiovascular pathologies, including hypertension, early onset of age-related decline in functions is observed [43]. This is characterized by progressive pathological remodeling with stiffening in the vascular system [44]. Vascular stiffening leads to widening arterial pulse pressure, aggravating endothelial dysfunction and vascular disease.

### 2.2. Microvascular Diseases in Diabetes

The microvascular disorder leads to retinopathy, nephropathy, and neuropathy, which, significantly, cause blindness, renal failure, and nerve injuries in patients with diabetes. Diabetic complications are initiated by to chronic hyperglycemia, which has an adverse effect on vascular tissues, including increased polyol pathway, increased diacylglycerol (DAG), activation of protein C kinase pathway, and increased the oxidative stress-increased hexosamine pathway, and action of advanced glycation end-products [45]. Worldwide diabetic retinopathy is responsible for 10,000 cases of blindness every year [46].

#### Pathophysiological Features of Microvascular Disease

Pathognomonic signs of diabetic microangiopathy include thickening of the capillary basement membrane, increased endothelial permeability, and dysfunction of endothelial and vascular smooth muscle cells [47]. Hyperglycemia is recognized as the primary factor for developing diabetic microvascular diseases. It was observed that pathologies in the retina and renal glomeruli are specific to diabetes and not present in elderly or insulin-resistant people without diabetes. In addition, hyperglycemia stimulates vasoinjurious signaling pathways [48], the polyol pathway, increases oxidative stress, promotes pro-inflammatory transcription factors, and triggers immune responses. Along with this, similar processes are persuaded by hypertension [49]. In the polyol pathway, intracellular glucose is converted to sorbitol by aldose reductase (AR), a rate-limiting enzyme, in a nicotinamide adenine dinucleotide phosphate (NADPH)-dependent reaction. Sorbitol is then oxidized to fructose by sorbitol dehydrogenase (SDH). In diabetes, increased intracellular glucose levels can cause increased flux through AR [50,51]. Again, activating the polyol pathway has been recommended to cause vascular pathologies by osmotic damage and reduce Na^+^-K^+^ ATPase activity [52]. 

Intracellular signaling molecules such as DAG and PKC can regulate many vascular functions. Receptor-mediated physiological PKC activation is mediated mainly by the activation of phospholipase C, which leads to an increase in Ca^2+^ and DAG levels [53]. Intracellular hyperglycemia increases glycolytic pathway flux and leads to an elevation of glycolytic intermediate dihydroxyacetone phosphate. Again, an increased level of this intermediate stimulates increases in the de novo synthesis of DAG [54]. Numerous studies have disclosed that in diabetes, DAG levels are increased in various tissues, such as the retina, glomeruli, aorta, and heart [50,55]. These persistent elevations of DAG levels lead to increased PKC activation, which is associated with alterations in blood flow, basement membrane thickening, ECM expansion, increased vascular permeability, abnormal angiogenesis, excessive apoptosis, increased leukocyte adhesion, and changes in enzymatic activity alterations, such as Na^+^-K^+^ ATPase, cytosolic phospholipase A_2_ (cPLA_2_), PI3K, and mitogen-activated protein kinase (MAPK) [56].

## 3. Interacting Mechanisms in T2DM and Hypertension

### 3.1. AGE–RAGE Axis

AGEs are oxidative derivatives resulting from diabetic hyperglycemia and are considered a potential risk for islet β-cell injury, peripheral insulin resistance, and diabetes [57]. Irreversible posttranslational modifications occur in advanced glycation end products (AGEs) because of reactions on proteins and nucleic acids between sugars and amino groups [58]. Hyperglycemia accelerates the formation of AGEs, which accumulate in the extracellular matrix of vessels and contribute to vascular damage in diabetes [27,59]. In addition to this, AGEs stimulate the production of reactive oxygen species (ROS), further increasing AGE formation. It has also been found that AGEs can induce immune responses as they are antigenic [60].

AGEs interact with two main types of cell surface receptors: (1) scavenger receptors and (2) receptors for AGEs (RAGE) [61,62]. AGE–RAGE signals through several factors, such as transforming growth factor (TGF)-β, NF-κB, mitogen-activated protein kinase, nicotinic adenine dinucleotide phosphate (NADPH) oxidases (Nox), and induces expression of vascular adhesion molecule 1, E-selectin, vascular endothelial growth factor, and pro-inflammatory cytokines such as IL-1β, IL-6, and TNF-α [63]. In diabetes, there is increased activation of these signaling pathways, particularly in vascular smooth muscle cells resulting in vascular fibrosis, calcification, inflammation, prothrombotic effects, and vascular damage [64,65]. In addition, these are the underlying processes for developing diabetic nephropathy, retinopathy, neuropathy, and atherosclerotic CVD [66]. These complications are increased in diabetes with coexisting hypertension [67]. It has been observed that patients with diabetes have increased tissue and circulating concentrations of AGEs and soluble RAGE, which is predictive of cardiovascular-related abnormal events and all-cause of mortality [68]. Targeting AGE–RAGE has been considered a potential therapeutic strategy to reduce or prevent CVD in diabetes [69].

### 3.2. Oxidative Stress and Nox

Endothelial cells regulate their vascular tone by releasing contracting and relaxing factors such as nitric oxide (NO), arachidonic acid metabolites, reactive oxygen species (ROS), and vasoactive peptides [70]. Oxidative stress is considered the primary mechanism for developing glucotoxicity in diabetes [71,72]. Moreover, hyperglycemia causes the generation of increased vascular ROS and the accumulation of oxidation by-products of lipids, proteins, and nucleic acids [73]. Oxidative stress is also associated with reduced bioavailability of the vasodilator nitric oxide resulting in endothelial dysfunction (Figure 2) [74]. NADPH oxidases (Nox) and dysfunctional endothelial nitric oxide synthase are significant sources of increased ROS in human vasculature in T2DM [75]. ROS interacts with DNA and stimulates many redox-sensitive signaling pathways that cause inflammation, fibrosis, and vascular damage [76]. Increased vascular oxidative stress in diabetes and hypertension promotes posttranslational oxidative modification of proteins, triggering cellular damage and vascular dysfunction [77].

### 3.3. Role of Inflammation and the Immune System

Immune cell infiltration is an important feature linking obesity to diabetes, as pro-inflammatory cytokines, macrophages, and T cells are essential for developing insulin resistance [78]. Moreover, Classical T cell activation is accompanied by upregulation of the insulin receptor, with a subsequent increase in Glut1, Glut3, and Glut4 and upregulation of glycolytic enzymes [79]. In addition, silencing the insulin receptor impairs T cell functions related to glucose transport and glycolysis, including polyclonal activation of CD4+ T cells, effector cytokine production, migration, and proliferation [80]. 

## 4. Insulin Resistance

Insulin resistance is defined as an impaired biologic response to insulin stimulation in target tissues, primarily the liver, muscle, and adipose tissue [81]. Insulin resistance impairs glucose removal, resulting in a compensatory increase in beta-cell insulin production and hyperinsulinemia [82]. Insulin resistance can be assessed by (i) HOMA-IR (Homeostasis Model Assessment of Insulin Resistance): calculated as the insulin level in mIU/L times glucose in mg/dl, divided by 405; and (ii) the Matsuda index as a measure of whole-body insulin resistance, calculated as 10,000 divided by the square root of FPG × fasting immunoreactive insulin (IRI) × 2 h post-load glucose × 2 h post load IRI. HOMA-IR is estimated using the homeostasis model assessment, which represents hepatic insulin resistance, and the Matsuda index reflects insulin resistance in the whole body, including skeletal muscle [83,84]. 

### 4.1. Insulin Resistance and Hypertension

Insulin is a pleiotropic hormone, and it regulates different physiological processes on glucose, lipid and protein metabolism, ion and amino acid transport, cell cycle, proliferation and differentiation, and nitric oxide (NO) synthesis [85]. In the vascular system, insulin stimulation persuades vasodilation through NO production, and during insulin resistance, there is impairment of NO synthesis, which causes altered vascular function [86]. In addition, cardiometabolic syndrome (CMS) and obesity are usually characterized by metabolic insulin resistance [87]. Under physiological conditions, insulin regulates glucose homeostasis by removing glucose in insulin-sensitive tissues and controlling nutrient supply through its vasodilation activity in small feed arteries [88]. Specifically, insulin-mediated nitric oxide production (NO) from vascular endothelium leads to increased blood flow, enhancing the removal process of glucose [89]. A recent study showed that increased plasma levels of insulin and aldosterone in states of insulin resistance lead to reduced bioavailable nitric oxide, causing impaired vascular relaxation and pathological vascular stiffening [90]. Furthermore, serum and glucocorticoid kinase 1 (SGK-1) is a significant regulator of vascular and renal sodium (Na^+^) channel activity, and this SGK-1 activity is increased by both insulin and aldosterone [91]. The study also revealed that mutations in SGK-1 in humans promote hypertension, insulin resistance, and obesity [92].

The relationship between insulin resistance and hypertension is a complex and multifactorial phenomenon that comprises a genetic basis and environmental factors [93]. It was observed that people in western countries have a sedentary lifestyle and consume hypercaloric food, which plays a significant role in developing insulin resistance, mainly through epigenetic modifications [94]. In addition, DNA methylation, histone modifications, and noncoding RNA activity (miRNA) are the chief mechanisms that alter protein transcription and expression and modify the cellular phenotype [95]. The translocation of GLUT4 to the cell membrane is the primary step of insulin-induced glucose uptake [96]. The insulin resistance state is characterized by lower expression levels and impaired translocation of GLUT4 [97]. Experimental data indicate that the methylation of DNA, induced by overnutrition during fetal life, decreases the gene expression of proteins involved in insulin signal transduction, such as GLUT4 [98]. In addition, GLUT4 is also affected by miRNA [99]. In myocytes, the miRNA 106b impairs insulin signaling by decreasing insulin-stimulated translocation of GLUT4 [100]. Mitochondrial dysfunction plays an essential role in the genesis of insulin resistance and is affected by epigenetic modifications [101]. Furthermore, methylation of the gene encoding for peroxisome proliferator-activated receptor alpha (PPARα) has been reported in obese subjects [102].

### 4.2. Blood Pressure and Insulin Resistance

It has been observed that insulin action is specific in spontaneously hypertensive rats (SHR) and is not related to compensatory hyperinsulinemia or hyperglycemia [103]. Conversely, in rats fed for 6 months with a hypercaloric diet, the increase in blood pressure and development of LVF is associated with either hyperinsulinemia or hyperglycemia [104]. Along with this, insulin resistance and the resultant hyperinsulinemia are responsible for developing hypertension-related target organ damage (TOD) through the defects of the counter-regulatory effects of insulin [105]. 

### 4.3. Molecular Mechanism of Insulin Resistance

Abnormalities of insulin signaling are responsible for insulin resistance. Insulin exerts its known physiological effects by binding to the insulin receptor on the plasma membrane of target cells [106]. The insulin receptor is a heterotetrameric receptor tyrosine kinase consisting of two extracellular α subunits, which bind insulin and two membrane-spanning β subunits, each containing a tyrosine kinase domain [107]. Binding insulin to the α subunit of its receptor activates the tyrosine kinase of the β subunit of the receptor, leading to autophosphorylation and tyrosine phosphorylation of several insulin receptor substrates (IRS), including IRS-1 and IRS-2. These, in turn, interact with phosphatidylinositol 3-kinase (PI3K) [107,108]. Activation of PI3K stimulates the main downstream effector AKT, a serine/threonine kinase, which enables the glucose uptake through the translocation of the major glucose transporter GLUT4 to the plasma membrane [109].

These insulin-signaling events, insulin receptor activation, and the phosphorylation of signaling proteins, prominently IRS, PI3K, and AKT isoforms, are mainly conserved in insulin target tissues and initiate the insulin response at the plasma membrane [110]. Insulin receptor defective function may contribute to insulin resistance, including abnormalities in receptor structure, number, binding affinity, and signaling capacity [111]. It has been suggested that hyperglycemia causes the progression of insulin resistance through the generation of reactive oxygen species (ROS), which abolish insulin-induced tyrosine autophosphorylation of the insulin receptor [112]. In developing hypertension, insulin receptor (IR) or insulin receptor substrate (IRS) signaling has a mechanistic role independent of glucose homeostasis and plasma insulin levels, indicating that insulin resistance is involved in the pathogenesis of hypertension [106,113].

### 4.4. Neuro-Hormonal Activities and Insulin Resistance

Several pathophysiological mechanisms impair insulin signaling in hypertension, such as renin–angiotensin, sympathetic nervous systems, and oxidative stress [114]. In addition, there is impairment of mechanisms that play a defensive role against insulin resistance in hypertension. The renin–angiotensin–aldosterone system (RAAS) plays a vital role in the pathogenesis of IR. Angiotensin II (Ang II), through the generation of reactive oxygen species (ROS), induces proteasome-mediated degradation of insulin receptor substrate-1 (IRS1), resulting in the impairment of insulin action [108]. This effect persuades a low-grade inflammation at the vascular level, which accounts for the development of insulin resistance and subsequent CV events [115].

### 4.5. Role of Prorenin

In diabetes, there is a marked increase in prorenin/renin levels, which may contribute to the development of diabetic nephropathy via interaction with the renal PRR [116]. Moreover, there is indirect evidence for enhanced renal medullary PRR expression in insulin resistance [117]. Researchers showed that mice with a null mutation of the carcinoembryonic antigen-related cell adhesion molecule 1 (Ceacam 1) had insulin resistance, visceral obesity, and postprandial hyperglycemia associated with increased expression of medullary PRR and activation of tubular RAS components [118]. Additionally, the same group had augmented renal medullary PRR expression and Ang-II levels, leading to hypertension after high fat intake [119]. Thus, renal PRR may have a role in the development and progression of renal injury in diabetes and insulin resistance.

### 4.6. Sympathetic Nervous System

The metabolic effects of insulin resistance, including hyperglycemia and dyslipidemia, appear to interact synergistically with increased BP to cause vascular and kidney injury that can exacerbate hypertension and associated damage to the kidneys and cardiovascular system [120]. It has been documented that some conditions of insulin resistance are characterized by up-regulation of the sympathetic nervous system, resulting in enhanced stimulation of β-adrenergic receptors (βAR) [121]. This increased stimulation of βAR causes heart failure and is related to insulin resistance in the heart [122].

## 5. Target Organ Damage in Insulin Resistance and Hypertension

Hypertension induces several manifestations of target organ damage: left ventricular hypertrophy (LVF), carotid atherosclerosis (CA), and renal dysfunction (Figure 3) [123]. In addition to this, insulin resistance stimulates the progress of left ventricular hypertrophy, carotid atherosclerosis, and renal chronic kidney disease (CKD) [124]. 

### 5.1. Left Ventricular Hypertrophy

LVF is a multidimensional process that involves genetic, hemodynamic, and anthropometric components, neurohormonal stimulation, growth factors, and inflammatory mediators [125,126]. The hemodynamic and metabolic disorders associated with insulin resistance increase the risk of LVF [127]. The growth factor actions of insulin, acting via insulin-like growth factor (IGF-1), may have an exaggerated hypertrophic response of the left ventricle to arterial hypertension [128]. IGF-1 and insulin signaling receptors share common intracellular protein substrates, and IGF-1 and insulin activate similar downstream signaling molecules, such as mitogen-activated protein kinase (MAPK) and PI3K [129]. IGF-1 receptor (IGF-1R) and insulin receptor (IR) have significantly different affinities to their cognate ligand despite having similarities [130]. While insulin resistance and hypertension coexist in an individual, a mixed pattern of cardiac hypertrophy occurs, caused by an elevation in preload and afterload [131]. Again, the myocardium in insulin-resistant hypertensive individuals has mononuclear cell infiltration and the conduction system, making the myocardium an ideal substrate for cardiac arrhythmia and sudden death [132]. 

### 5.2. Carotid Atherosclerosis

Several metabolic alterations persuade the progression of cardiovascular disease during an insulin-resistant state. One study reveals insulin resistance damages the myocardium by signal transduction alteration, impaired substrate metabolism regulation, and altered substrate delivery to the myocardium [133]. Under physiological conditions, insulin stimulates metabolic substrates in multiple tissues, including the heart, skeletal muscle, liver, and adipose tissue. Insulin promotes glucose and fatty acid uptake in cardiomyocytes, but inhibits fatty acids as an energy source. As a result of insulin resistance, compensatory hyperinsulinemia occurs [134]. Hypertensive patients with insulin resistance have impaired vasodilation, which results in a decrease in peripheral blood flow.

Moreover, these patients have more intima-media thickness (IMT) in the common carotid artery (CCA) than those without insulin resistance (IR) [135]. Another study demonstrated that hypertensive patients with IR have a high prevalence of producing plaques and increased arterial stiffness of CCA in comparison with hypertensive subjects without IR. This study also showed that endothelial dysfunction is related to disturbance production in endothelial-dependent vasodilation in IR. The CCA significantly decreased blood flow velocity and relative diastolic blood flow in the hypertensive patients in the insulin resistance group [136]. 

### 5.3. Renal Dysfunction

Chronic kidney disease is defined as a progressive glomerular, tubular, and interstitial injury with loss of nephron function due to glomerular sclerosis and tubular atrophy [137]. Diabetes, hypertension, and insulin resistance are the leading cause of chronic kidney disease (CKD), and frequently develop into end-stage renal disease (ESRD) [138]. Hypertension causes CKD by increasing glomerular capillary pressure, proteinuria, and endothelial dysfunction leading to nephron damage [139]. Insulin resistance is associated with stimulating both the renin–angiotensin system and sympathetic system activities, contributing to increased renal sodium reabsorption, fluid retention, and hypertension [140]. Furthermore, there is an increased proliferation of endothelial cells and deposition of intrarenal lipid hyaluronate in the matrix and inner medulla [141]. These depositions increase intrarenal pressure and volume in the tightly encapsulated kidney, resulting in parenchymal prolapse and urine outflow obstruction, leading to slow tubular flow and subsequently increased sodium reabsorption in the loop of Henle. Thus, impaired pressure natriuresis occurs. At the same time, due to these structural and functional changes, there is an augmentation of compensatory lowered renal vascular resistance, increased renal plasma flow, glomerular hyperfiltration, and stimulation of the renin–angiotensin system despite volume expansion. In addition, these changes raised blood pressure with insulin resistance, increased tubular reabsorption, and maintained sodium balance. The persistence of these compensatory responses eventually leads to increased glomerular wall stress, gradual nephron loss and glomerulosclerosis, and ultimately end-stage renal disease.

## 6. Conclusions and Recommendation

Diabetes leads to the accelerated generation of advanced glycation end products (AGEs) and activation of their receptor, RAGE, and NADPH oxidase (Nox), leading to a pro-inflammatory environment characterized by oxidative stress. A significant shift over the past half-century is the enrichment of the food environment with AGEs, appetizing pro-oxidant substances, which can promote both overnutrition and oxidant overload. Continuous oxidant overload may overwhelm host defenses and lead to unopposed oxidant stress and chronic inflammation. These states can, over time, impair insulin production and sensitivity and lead to diabetes. We focused on vascular complications of diabetes and discussed the impact of co-morbidities, specifically hypertension. The role of oxidative stress and inflammation as “common soil” for metabolic and vascular disease are highlighted. Controlling co-morbidities, such as hypertension, and targeting strategies to promote vascular health, may be especially important in reducing the microvascular and macrovascular complications of diabetes. Understanding insulin action and resistance more completely will facilitate the intelligent use of existing anti-diabetic therapies, enable the development of new therapeutics, and perhaps most importantly, inform prevention strategies to stem the tide of type 2 diabetes. 

The risk of morbidity and mortality from cardiovascular diseases increases intensely when blood pressure is not well controlled in diabetes mellitus and hypertension individuals. Blood pressure control is an essential part of managing patients with diabetes as it is one of the most effective ways to prevent vascular complications and death. Patients with T2DM have increased tissue and circulating concentrations of AGEs and soluble RAGE, which can predict cardiovascular events and similar risk factors related to mortality. Targeting AGE–RAGE has been considered a potential therapeutic strategy to reduce or prevent CVD in diabetes. In addition, urinary and plasma AGE levels and soluble RAGE may act as biomarkers for vascular disease in diabetes. Insulin resistance needs to be considered the main target of therapeutic strategies designed to reduce cardio-metabolic risk. This goal can be achieved by combining dietary changes, regular physical exercise, and the pharmacological inhibition of RAAS.

## Figures and Tables

**Figure 1 life-12-00564-f001:**
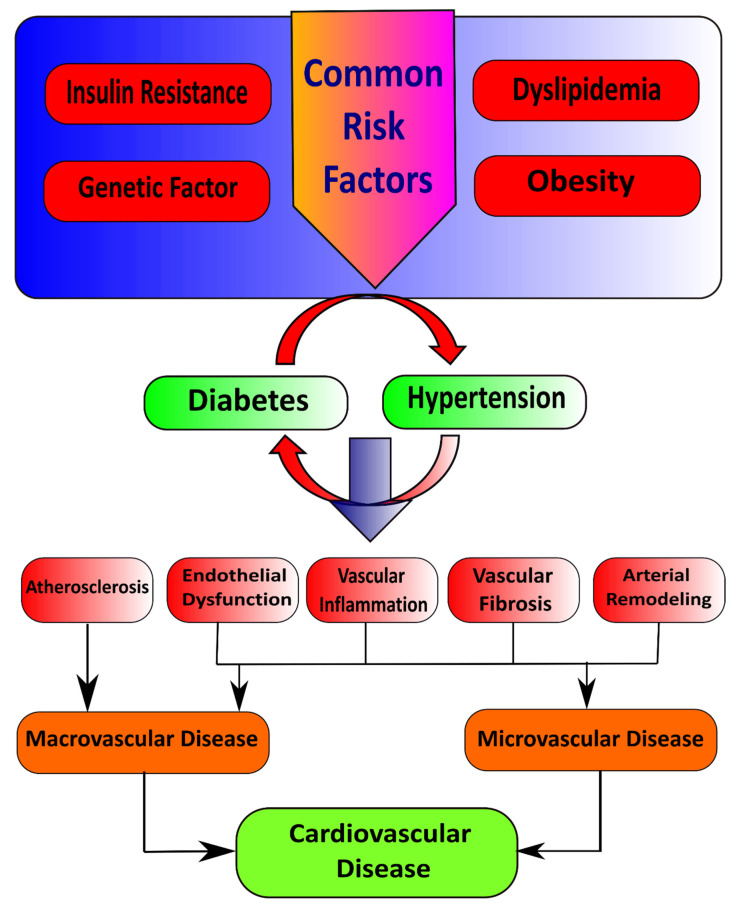
Illustration of common risk factors that cause diabetes and hypertension.

**Figure 2 life-12-00564-f002:**
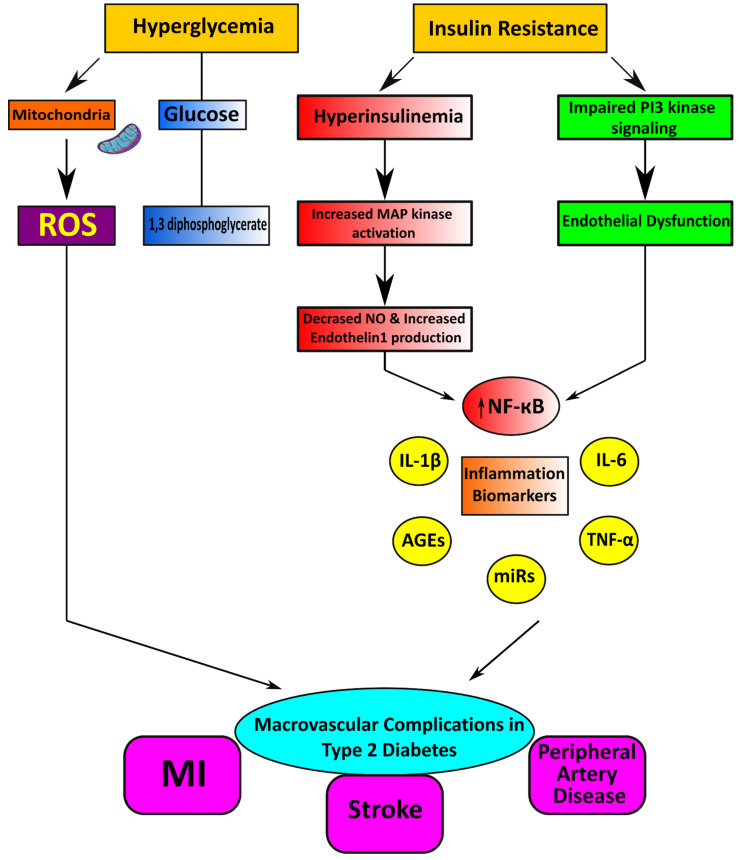
Cardiometabolic disorders result from vascular complications in type 2 diabetes mellitus.

**Figure 3 life-12-00564-f003:**
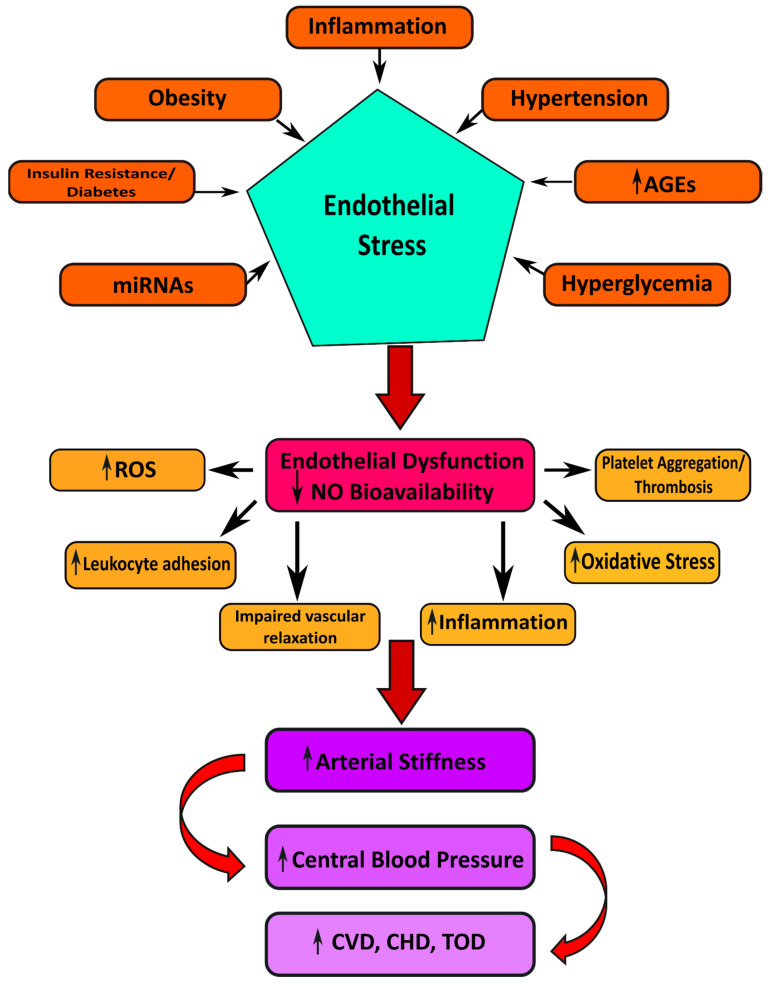
Illustrating the possible mechanisms linked to the pathogenesis of cardiovascular manifestations in Type 2 diabetes mellitus.

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
