# Peer review of "Insulin Resistance Is Cheerfully Hitched with Hypertension"

_life, 2022, doi:10.3390/life12040564_

Round 1

Reviewer 1 Report

Intersting article higlighting the role of renin-angiotensin aldosterone system, oxidative stress, inflammation, and immune system,
in diabetes and hypertension association. Based on the available literatures, authors have concluded significant increase type 2 diabetes risk in the indivduals having above-mentioned issues. My remarks are mentioned based on the star marking. I recommend minor revision for this article 

Author Response

Reviewer I

Open Review

() I would not like to sign my review report.

(x) I would like to sign my review report

English language and style

() Extensive editing of English language and style required.

(x) Moderate English changes required.

() English language and style are fine/minor spell check required.

() I don't feel qualified to judge about the English language and style.

Is the work a significant contribution to the field?

Is the work well organized and comprehensively described?

Is the work scientifically sound and not misleading?

Are there appropriate and adequate references to related and previous work?

Is the English used correct and readable?

Comments and Suggestions for Authors

Interesting article highlighting the role of renin-angiotensin aldosterone system, oxidative stress, inflammation, and immune system, in diabetes and hypertension association. Based on the available literatures, authors have concluded significant increase type 2 diabetes risk in the individuals having above-mentioned issues. My remarks are mentioned based on the star marking. I recommend minor revision for this article 

Thanks Sis for your kind comment. We have taken care of English Language.

Submission Date

15 March 2022

Date of this review

30 Mar 2022 12:44:25

Reviewer II

Open Review

(x) I would not like to sign my review report.

() I would like to sign my review report.

English language and style

() Extensive editing of English language and style required.

() Moderate English changes required.

(x) English language and style are fine/minor spell check required.

() I don't feel qualified to judge about the English language and style

Is the work a significant contribution to the field?

Is the work well organized and comprehensively described?

Is the work scientifically sound and not misleading?

Are there appropriate and adequate references to related and previous work?

Is the English used correct and readable?

Comments and Suggestions for Authors

The authors outline the role of hypertension in type 2 diabetes progression and the relation between them. While the manuscript has several strengths, there are few issues that need to be addressed, mainly related to some spelling observations:

  1. Using sections like: Aim and Materials and Methods is not very common in Review Articles. I would recommend skipping them. Also, you should consider including the Recommendation as final paragraph in the Conclusion section.

Dear Sir, earlier we experienced and need to add Materials and Methods. They referred this paper https://researchintegrityjournal.biomedcentral.com/articles/10.1186/s41073-019-0064-8. Thereby, we add and desire to retain. Moreover, REVIEWER II does not have any objection.

  1. For a better understanding I would recommend several spell checking:

-  line 111: “target” becomes “targeted”

-  line 116: replace “related to “with “associated with”

-  line 116: replace “conduit vessel” with “conduits”

-  line 116: place “and” between “macrovascular” and “microvascular”

-  line 117: better “Chronic hyperglycemia and insulin resistance have an important role in the initiation of vascular complications of diabetes consisting in several mechanisms, such as:……”

- line 136: better “Adipocytes, in both subcutaneous ad visceral areas, undergo hypertrophy during calorie excess, in obese humans.”

- line 140: lose “expression”

- line 154: replace “disease” with “disorder”

- line 156: replace “due to “with “by”

- do not use so much “It” instead of the suitable noun, may be confusing (line 135: maybe “insulin resistance” and line 159: maybe “diabetes retinopathy”)

- line 162: replace “changes in” with “signs of”

- line 258: replace “observed” with “showed”

- line 262: replace “they also observed” with “the study also revealed”

- line 310: replace “it specifies” with “indicating”

- line 325: replace “They” with “Researchers” and you also missed “that” after “showed”

- line 408: “focus” becomes “focused” and “discuss” becomes “discussed”

Thanks Sir. We have corrected point to point as per you instructed.

Overall, the paper needs minor editing.

 Submission Date

15 March 2022

Date of this review

28 Mar 2022 09:05:58

Reviewer III

Open Review

() I would not like to sign my review report.

(x) I would like to sign my review report

English language and style

() Extensive editing of English language and style required.

() Moderate English changes are required.

(x) English language and style are fine/minor spell check required.

() I don't feel qualified to judge about the English language and style

Is the work a significant contribution to the field?

Is the work well organized and comprehensively described?

Is the work scientifically sound and not misleading?

Are there appropriate and adequate references to related and previous work?

Is the English used correct and readable?

Comments and Suggestions for Authors

This is a spectacular review of the relation of IR to hypertension and related disorders, e.g., CVD. It will be of great value to basic scientists and clinicians. I give it my highest recommendation and have no substantive suggestions, other than I'm not really happy with the title. It seems to me to be more of an unhappy marriage than a happy one, since the marriage leads to so much grief. Perhaps the authors could rethink the title.

Thanks, Sir for your kind comment. We have taken care of the English Language. And altered the title.

Submission Date

15 March 2022

Date of this review

05 Apr 2022 19:55:31

Reviewer 2 Report

The authors outline the role of hypertension in type 2 diabetes progression and the relation between them. While the manuscript has several strengths, there are few issues that need to be addressed, mainly related to some spelling observations:

  1. Using sections like: Aim and Matherials and Methods is not very common in Review Articles. I would recommend skipping them. Also, you should consider including the Recommendation as final paragraph in the Conclusion section.
  2. For a better understanding I would recommend several spell checking:

-  line 111: “target” becomes “targeted”

-  line 116: replace “related to“ with “associated with”

-  line 116: replace “conduit vessel” with “conduits”

-  line 116: place “and” between “macrovascular” and “microvascular”

-  line 117: better “Chronic hyperglycemia and insulin resistance have an important role in the initiation of vascular complications of diabetes consisting in several mechanisms, such as:……”

- line 136: better  “Adipocytes, in both subcutaneous ad visceral areas, undergo hypertrophy during calorie excess, in obese humans.”

- line 140: lose “expression”

- line 154: replace “disease” with “disorder”

- line 156: replace “due to“ with “by”

- do not use so much “It” instead of the suitable noun, may be confusing (line 135: maybe “insulin resistance” and line 159: maybe “diabetes retinopathy”)

- line 162: replace “changes in” with “signs of”

- line 258: replace “observed” with “showed”

- line 262: replace “they also observed” with “the study also revealed”

- line 310: replace “it specifies” with “indicating”

- line 325: replace “They” with “Researchers” and you also missed  “that” after “showed”

- line 408: “focus” becomes “focused” and “discuss” becomes “discussed”

Overall, the paper needs minor editing.

Author Response

Reviewer I

Open Review

() I would not like to sign my review report.

(x) I would like to sign my review report

English language and style

() Extensive editing of English language and style required.

(x) Moderate English changes required.

() English language and style are fine/minor spell check required.

() I don't feel qualified to judge about the English language and style.

Is the work a significant contribution to the field?

Is the work well organized and comprehensively described?

Is the work scientifically sound and not misleading?

Are there appropriate and adequate references to related and previous work?

Is the English used correct and readable?

Comments and Suggestions for Authors

Interesting article highlighting the role of renin-angiotensin aldosterone system, oxidative stress, inflammation, and immune system, in diabetes and hypertension association. Based on the available literatures, authors have concluded significant increase type 2 diabetes risk in the individuals having above-mentioned issues. My remarks are mentioned based on the star marking. I recommend minor revision for this article 

Thanks Sir for your kind comment. We have taken care of English Language.

Submission Date

15 March 2022

Date of this review

30 Mar 2022 12:44:25

Reviewer II

Open Review

(x) I would not like to sign my review report.

() I would like to sign my review report.

English language and style

() Extensive editing of English language and style required.

() Moderate English changes required.

(x) English language and style are fine/minor spell check required.

() I don't feel qualified to judge about the English language and style

Is the work a significant contribution to the field?

Is the work well organized and comprehensively described?

Is the work scientifically sound and not misleading?

Are there appropriate and adequate references to related and previous work?

Is the English used correct and readable?

Comments and Suggestions for Authors

The authors outline the role of hypertension in type 2 diabetes progression and the relation between them. While the manuscript has several strengths, there are few issues that need to be addressed, mainly related to some spelling observations:

  1. Using sections like: Aim and Materials and Methods is not very common in Review Articles. I would recommend skipping them. Also, you should consider including the Recommendation as final paragraph in the Conclusion section.

Dear Sir, earlier we experienced and need to add Materials and Methods. They referred this paper https://researchintegrityjournal.biomedcentral.com/articles/10.1186/s41073-019-0064-8. Thereby, we add and desire to retain. Moreover, REVIEWER II does not have any objection.

  1. For a better understanding I would recommend several spell checking:

-  line 111: “target” becomes “targeted”

-  line 116: replace “related to “with “associated with”

-  line 116: replace “conduit vessel” with “conduits”

-  line 116: place “and” between “macrovascular” and “microvascular”

-  line 117: better “Chronic hyperglycemia and insulin resistance have an important role in the initiation of vascular complications of diabetes consisting in several mechanisms, such as:……”

- line 136: better “Adipocytes, in both subcutaneous ad visceral areas, undergo hypertrophy during calorie excess, in obese humans.”

- line 140: lose “expression”

- line 154: replace “disease” with “disorder”

- line 156: replace “due to “with “by”

- do not use so much “It” instead of the suitable noun, may be confusing (line 135: maybe “insulin resistance” and line 159: maybe “diabetes retinopathy”)

- line 162: replace “changes in” with “signs of”

- line 258: replace “observed” with “showed”

- line 262: replace “they also observed” with “the study also revealed”

- line 310: replace “it specifies” with “indicating”

- line 325: replace “They” with “Researchers” and you also missed “that” after “showed”

- line 408: “focus” becomes “focused” and “discuss” becomes “discussed”

Thanks Sir. We have corrected point to point as per you instructed.

Overall, the paper needs minor editing.

 Submission Date

15 March 2022

Date of this review

28 Mar 2022 09:05:58

Reviewer III

Open Review

() I would not like to sign my review report.

(x) I would like to sign my review report

English language and style

() Extensive editing of English language and style required.

() Moderate English changes required.

(x) English language and style are fine/minor spell check required.

() I don't feel qualified to judge about the English language and style

Is the work a significant contribution to the field?

Is the work well organized and comprehensively described?

Is the work scientifically sound and not misleading?

Are there appropriate and adequate references to related and previous work?

Is the English used correct and readable?

Comments and Suggestions for Authors

This is a spectacular review of the relation of IR to hypertension and related disorders, e.g., CVD. It will be of great value to basic scientists and clinicians. I give it my highest recommendation and have no substantive suggestions, other than I'm not really happy with the title. It seems to me to be more of an unhappy marriage than a happy one, since the marriage leads to so much grief. Perhaps the authors could rethink the title.

Thanks Sir for your kind comment. We have taken care of English Language. And altered the title.

Submission Date

15 March 2022

Date of this review

05 Apr 2022 19:55:31

Reviewer 3 Report

This is a spectacular review of the relation of IR to hypertension and related disorders, e.g., CVD. It will be of great value to basic scientists and clinicians. I give it my highest recommendation and have no substantive suggestions, other than I'm not real happy with the title. It seems to me to be more of an unhappy marriage than a happy one, since the marriage leads to so much grief. Perhaps the authors could rethink the title.

Author Response

(The authors gave the same response as above.)
